# Association of Physical Fitness with Intelligence and Academic Achievement in Adolescents

**DOI:** 10.3390/ijerph17124362

**Published:** 2020-06-18

**Authors:** Francisco Javier Gil-Espinosa, Palma Chillón, José Carlos Fernández-García, Cristina Cadenas-Sanchez

**Affiliations:** 1Department of Didactics of Languages, Arts and Sport, University of Malaga, Andalusia-tech, 29071 Málaga, Spain; jcfg@uma.es; 2PROFITH “PROmoting FITness and Health through Physical Activity” Research Group, Department of Physical and Sports Education, Faculty of Sport Sciences, University of Granada, 18071 Granada, Spain; pchillon@ugr.es (P.C.); cadenas@ugr.es (C.C.-S.); 3MOVE-IT research group, Department of Physical Education, Faculty of Education Sciences University of Cádiz, 11519 Cádiz, Spain; 4Institute for Innovation & Sustainable Development in Food Chain (IS-FOOD), Public University of Navarra, 31006 Pamplona, Spain

**Keywords:** cardiorespiratory fitness, flexibility, strength, cognition, academic performance, youth

## Abstract

Physical fitness, intelligence and academic achievement are being studied from a multidisciplinary perspective. In this line, studies to advance our understanding of intelligence and academic achievement could be relevant for designing school-based programs. Our study analyzed the relationship between components of physical fitness including cardiorespiratory fitness, muscular strength and flexibility and general intelligence and academic achievement in adolescents. We recruited 403 adolescents (53.6% boys) with a mean age of 13.7 ± 1.2 years from a secondary school in Spain with a medium socioeconomic status, during the 2015/2016 school year. Cardiorespiratory fitness was assessed by the 20-m shuttle run, muscular strength with the standing long jump test and flexibility with the sit-and-reach test. General intelligence was measured by both the D48 and the Raven tests. School grades were used to determine academic achievement. Linear regression analyses showed that cardiorespiratory fitness was positively associated with intelligence in both the D48 (all β ≥ 0.184, *p* ≤ 0.016) and the Raven tests (all β ≥ 0.183, *p* ≤ 0.024). Muscular strength, flexibility and overall fitness were not associated with intelligence (all β ≤ 0.122, *p* ≥ 0.139). Cardiorespiratory fitness, muscular strength and flexibility were positively associated with academic achievement (all β ≥ 0.089, *p* ≤ 0.038), except muscular strength, which was not significantly associated with Spanish language or mathematics, (all β ≤ 0.050, *p* ≥ 0.200). Overall, cardiorespiratory fitness was positively associated with intelligence and academic achievement.

## 1. Introduction

In recent years, a growing multidisciplinary field of knowledge has identified cognitive and academic benefits of engaging in regular physical activity [1,2]. Physical fitness in general, as well as cardiorespiratory fitness in particular, is a powerful marker of physical and mental health [3,4]. A recent systematic review with more than 319,000 children and youths from 32 different countries concluded that cardiorespiratory fitness was favorably associated with adiposity and cardiometabolic, cognitive and psychosocial health in boys and girls [3].

Cognition is defined as a set of mental processes that contribute to perception, memory, intellect and action [5], which in turn are related and intertwined with intelligence. Thus, intelligence generally describes a cognitive skill to solve problems and understand concepts in which processing speed, executive control, working memory, reasoning, reflection and awareness are relevant components [6,7]. At a cross-sectional level, Scudder et al. [8] reported that more highly fit participants (classified according to their cardiorespiratory fitness level) had a higher but not significant intelligence quotient than those with a lower physical fitness level. However, the literature regarding physical fitness and intelligence in youth is scarce, and little is known about the relationship between them. Longitudinally, within the same cohort used in the present study, we concluded that general intelligence may not predict physical fitness, but it may predict better academic achievement in adolescents one year later [9]. However, we found very little information showing the association between physical fitness components and general intelligence in adolescents. The relationship is inconsistent in the literature, which could be associated with differences in populations and the measurement tests used.

The academic achievement of students in compulsory secondary schools is continuously evaluated, and undoubtedly many variables may be involved. The human being is multidimensional and, as a consequence, education should have a holistic vision of students [10]. Nonetheless, in recent years, many schools have decreased physical education sessions or physical education time in an attempt to provide more time for academic subjects [11,12] such as mathematics, language or geography and history. However, over the past few years, a growing body of literature has shown that this assumption is incorrect. Accordingly, in 2016, the American College of Sports Medicine issued a position paper on physical activity, fitness, cognition and academic achievement in children, concluding that the association between physical fitness and academic achievement demonstrates largely positive findings [5]. In this regard, Cadenas-Sanchez et al. [13] observed that physical fitness components other than cardiorespiratory fitness, such as muscular strength and speed-agility, are positively associated with academic achievement in children. In addition, another study from the same cohort showed that cardiorespiratory fitness, as well as muscular strength and speed-agility, are positively associated with gray matter volume in several brain regions, which in turn is related to higher academic achievement [14]. However, most of the available information focuses on cardiorespiratory fitness, and little is known about the other components of fitness and their relationship to academic achievement [15].

Therefore, through increased understanding of intelligence and academic achievement, new training programs could be designed and organized for schools. This assumption is highly relevant for low academic achievers whose academic achievement might benefit from enhanced physical fitness [16]. Thus, our study contributes to closing this research gap through an analysis of the relationships between physical fitness, general intelligence and academic achievement in adolescents.

## 2. Materials and Methods

### 2.1. Participants

This study was part of the “Cognitive performance, academic achievement and fitness: Longitudinal and cross-sectorial” project (reference number PIV-006/17). The aim of this project was to examine the relationship between physical fitness, cognition and academic achievement in adolescents from the south of Spain [9,17]. For this study, we used cross-sectional data.

We recruited a convenience sample of 403 adolescents (53.6% boys) with a mean age of 13.7 ± 1.2 years from a secondary school in Andalusia, Spain, to participate in our study. We performed statistical power calculations using the G*Power software (https://www.psychologie.hhu.de/arbeitsgruppen/allgemeine-psychologie-und-arbeitspsychologie/gpower.html) [18]. Our sample size is enough to detect small association sizes (f^2^ ≥ 0.02) considering a statistical power of 80% and an alpha error of 0.05. The socioeconomic status of the school was classified as medium based on the Andalusian Educational Evaluation Agency report (level five out of ten), according to items including family income, parents’ profession, parents’ educational level and facilities for studying at home. The inclusion criteria were (1) being enrolled in compulsory secondary education and (2) having complete and valid data on physical fitness and intelligence or academic achievement.

Data collection took place during the first term of the 2015/2016 school year and was approved by the educational administration and the school council (according to Spanish educational legislation on the implementation of research). In addition, the parents and legal guardians were informed of the aims of the project and provided consent for their child to participate. The study complied with the ethical considerations concerning Sport and Exercise Science Research [19] and the principles of the Declaration of Helsinki [20] (reference PIV-006/17). In addition, throughout the entire intervention and after it was completed, the project was conducted under the provisions of Organic Law 3/2018, of December 5, on the Protection of Personal Data and Guarantee of Digital Rights.

### 2.2. Measurements

#### 2.2.1. Physical Fitness

Physical fitness was measured using the EUROFIT fitness test battery [21] to determine cardiorespiratory fitness, muscular strength and flexibility. Assessments were done during physical education classes in two different days: first day, the 20 m shuttle run test was performed, and the second day we performed the standing long jump and sit-and-reach tests. For feasibility reasons, we did not apply the full EUROFIT fitness test battery.

Cardiorespiratory fitness was assessed by the 20 m shuttle run test, which involved running between two lines 20 m apart. The initial velocity was 8.5 km/h, which increased to 0.5 km/h per minute. The pace was marked by an audio signal. The test ended when the adolescents stopped due to fatigue or because they failed to reach the end line when the audio signal sounded. For the analyses, we recorded the stages achieved during the test. Then, the maximum oxygen consumption (mL/kg/min) was calculated following the Léger et al. [22] equation (Y = 31.025 + 3.238 × speed − 3.248 × age + 0.1536 × age × speed). This test was carried out once.

Muscular strength was evaluated by the standing long jump test, which measures explosive leg strength and is considered a general index of muscular fitness in this age group [23]. Briefly, the participant had to jump as far as possible, remaining upright and using both feet. The feet were positioned shoulder-width apart. The measurement was taken from the take-off line to the nearest point of contact on the landing. We recorded the best score, which was the longest distance jumped of the two attempts.

Flexibility was assessed with the sit-and-reach test which specifically measures the flexibility of the lower back and hamstring muscles. The participant was required to sit barefoot on the floor with legs fully extended and with feet flat against the box, then reach forward progressively along the measuring line as far as possible. We recorded in centimeters the furthest position reached on the two attempts.

A composite score of overall fitness was calculated using the mean of each sex-specific z-score obtained in each fitness component.

#### 2.2.2. General Intelligence

We assessed intelligence using two tests: the D48 and Raven’s Progressive Matrices.

The D48 test, also called the dominoes test, is a nonverbal test used to measure general intelligence [24] and is considered to have high internal consistency and reliability [24]. Moreover, it has been highly associated with general intelligence [25]. The D48 test contains 44 problems, each consisting of a series of dominoes that define a principle of progression. The last domino in the series is blank for the subject to complete. The purpose of this test is to identify and associate domino pieces. The test was supervised by a qualified person experienced in using this test in educational settings. Raw score was used for the analyses.

Raven’s Progressive Matrices tests measure general intelligence. They comprise a series of 60 diagrams or designs with a missing part, distributed in five series in order of increasing difficulty. Those taking the tests are instructed to select the correct piece to complete the designs from a number of options printed beneath each design [26]. Raw score was used for the analyses.

#### 2.2.3. Academic Achievement

Academic achievement was assessed by school grades. Specifically, we analyzed the school grades obtained in the first term of the assessment year. The subjects examined were Spanish language, mathematics, geography and history, English as a foreign language and physical education, in consonance with previous studies [27]. The grade point average was calculated based on the mean of these subjects. Likewise, the grade point average was calculated without including the grade in physical education, in line with previous studies [28,29].

#### 2.2.4. Statistical Analyses

Descriptive analyses were performed to display the characteristics of our study sample. To test differences between sexes, we used the independent *t*-test to determine whether there were differences in the mean between boys and girls. We also examined the interactions between sex and physical fitness and academic achievement or general intelligence, finding no evidence (*p* > 0.1). Confounders were selected after examining their influence on our outcomes and based on our previous research published in this same cohort [9,17]. Thus, data were adjusted for sex and age. The relationship between fitness components and academic achievement, and general intelligence was examined using linear regression analyses (standardized beta, unstandardized beta, 95% confidence interval and *p* value) adjusting for sex and age. All the linear regressions analyses were performed separately for each fitness component and each general intelligence and academic achievement outcome. Statistical significance was considered at *p* < 0.05.

Likewise, we divided the sample into unfit and fit groups based on cardiorespiratory fitness, muscular muscle strength, flexibility and overall fitness level measured according to the z-score for each fitness component. Participants were allocated to the unfit group when they had a z-score lower than 0, and those with a z-score of ≥0 were assigned to the fit group. Analyses of covariance (ANCOVA) adjusted for sex and age were performed to test differences between the unfit and fit groups in academic achievement and general intelligence.

All the analyses were performed using SPSS Statistics for Windows, version 20.00 (IBM Corp, Armonk, NY, USA).

## 3. Results

### 3.1. Descriptive Characteristics

The descriptive characteristics of the study sample (total, boys and girls) are displayed in Table 1. Briefly, differences between the sexes were found in cardiorespiratory fitness, muscular strength and flexibility (total *p* < 0.001). No significant difference was found between boys and girls on the D48 general intelligence test (*p* = 0.321). However, a significant difference was found on the Raven general intelligence test (*p* = 0.036). The girls showed higher achievement in Spanish language and English and had higher grade point averages than the boys (*p* < 0.02). To note that sample size decreases in some variables due to the missing data (i.e., did not attend to school the day of the evaluation). Specially, the general intelligence evaluation was decreased up to 161 participants, only including those participants that were engaged at the first level of the Spanish educational system.

### 3.2. Relationship between Physical Fitness Components and General Intelligence and Academic Achievement

Table 2 shows the associations between physical fitness components and general intelligence and academic achievement. Cardiorespiratory fitness was positively related to general intelligence in both the D48 (total β ≥ 0.184, *p* ≤ 0.016) and Raven tests (total β ≥ 0.183, *p* ≤ 0.024). Muscular strength, flexibility and overall fitness were not significantly associated with general intelligence (total β ≤ 0.122, *p* ≥ 0.139).

Cardiorespiratory fitness, muscular strength, flexibility and overall fitness were positively associated with academic achievement in all academic subjects (all β ≥ 0.089, *p* ≤ 0.038), except muscular strength, which was not significantly associated with Spanish language or mathematics (all β ≤ 0.050, *p* ≥ 0.200).

Figure 1 depicts the association between the unfit and fit groups and general intelligence. Briefly, adolescents categorized in the fit group for cardiorespiratory fitness had a higher general intelligence than those in the unfit group (total *p* < 0.005) on both general intelligence tests. However, the fit group did not have significantly better results than the unfit group for muscular strength, flexibility or overall fitness (total *p* > 0.05).

Figure 2 shows the relationship between the unfit and fit groups and academic achievement outcomes. Overall, students classified as fit presented better academic achievement than those in the unfit group (all *p* < 0.05).

## 4. Discussion

The main findings of our study suggest that cardiorespiratory fitness is associated with general intelligence. However, no significant difference was observed for muscular strength or flexibility. Further, cardiorespiratory fitness, muscular strength and flexibility are associated with academic achievement in adolescents. Specifically, the fitness components were positively associated with academic achievement, except muscular strength, which was not significantly associated with Spanish language or mathematics.

We observed that cardiorespiratory fitness was the only fitness component associated with intelligence. Our findings are in line with another study in which cardiovascular fitness is associated with intelligence in young adulthood [30]. Nonetheless, research focusing on the association between cardiorespiratory fitness and intelligence in youth is scarce. Most of the evidence published in children and adolescents addresses other cognitive domains such as executive function, memory and attention. In this regard, several cross-sectional studies found a positive association between higher levels of physical fitness and cognition in adolescents, in memory, attention and executive function [31,32]. An European study in adolescents showed that cardiorespiratory fitness (but not muscular strength or speed-agility) showed a positive association with attention [33]. The results provided by the European study in adolescents are in partial agreement with our findings. Although the cognitive indicator was not the same (i.e., HELENA study = attention versus our study = intelligence), the finding that having a good level of cardiorespiratory fitness is a marker of mental health is the same. Concerning physical fitness, the physiological response to a higher level of physical activity (and therefore better physical fitness) could explain why associations are found in cardiorespiratory fitness and not in other fitness components such as muscular strength or flexibility. Studies by Whiteman et al. [34] and Khan et al. [35] have recognized that high levels of physical activity promoted brain changes that can be explained by increased oxygenation and tissue irrigation, as well as increased metabolic activity, providing improved neurological development. Thus, the role of the intensity of the physical activity could also be a determining factor [36]. Similarly, other studies have indicated significantly improved brain function and behavioral performance after physical activity interventions [37,38]. People with higher intelligence tend to have better mental and physical health and fewer illnesses throughout life as well as longer lives [39]. Nevertheless, due to the limited number of studies published in this field, the different methods used for assessing intelligence and/or physical fitness and the unclear findings, we are unable to compare our findings with those of others. Consequently, although our results show a promising and strong relationship between cardiorespiratory fitness and intelligence, more studies in youth are needed in order to provide a clear message.

Regarding muscular strength and flexibility and their relationship with intelligence, our findings showed no significant association. However, to the best of our knowledge, no previous studies have examined muscular strength and flexibility in relation to intelligence. Cadenas-Sanchez et al. [33] examined the association between muscular strength and attention in European adolescents, concluding no significant association. Although different cognitive functions were measured, our findings are in accordance with those found in European adolescents. Consistent with our findings, a large cohort study of more than one million Swedish young adults [30] found that muscular strength was not related to intelligence achievement. Flexibility is a less studied component of fitness. Our findings showed no significant difference between flexibility and intelligence. We cannot compare our findings with those of others due to the lack of knowledge in the field of flexibility fitness and cognition in adolescents.

In recent years, a growing body of evidence has emerged on the relationship between physical fitness and academic achievement. Our findings showed that not only cardiorespiratory fitness but also muscular strength and flexibility were associated with higher academic achievement in adolescents. These results contribute to the existing literature supporting the idea that physical fitness plays a role in academic achievement in adolescents. There is strong evidence for a positive association between cardiorespiratory fitness and academic achievement [5,40]. Overall, the literature on flexibility and muscular strength is scarce in comparison with studies on cardiorespiratory fitness [15]. In line with our results, other studies found a positive association between cardiorespiratory fitness and academic achievement [41,42]. Similar to our findings, Cadenas-Sanchez et al., found that in preadolescent overweight-obese children cardiorespiratory fitness is associated with language components [13]. Moreover, the most well-known relationship examined is the association between cardiorespiratory fitness and mathematics [43,44]. Further studies in preadolescent children showed that children classified as fit based on cardiorespiratory fitness achievement showed decreased gray matter thickness in different areas of the brain coupled with better mathematic achievement compared to those classified as less fit [43]. However, another study found that fundamental motor skills were stronger predictors of academic achievement than aerobic fitness [16].

With regard to muscular strength, we found significant associations with geography and history, English, physical education and grade point average, but not with mathematics or Spanish Language. Though the scientific literature shows inconclusive results [15]. On the one hand, and corroborating our findings, Cadenas-Sanchez et al. [13] found that in preadolescent children muscular strength was associated with geography and history, English and grade point average. On the other hand, Bilgin et al. [42] found a positive relationship between muscular strength and academic achievement in boys but not in girls. Therefore, although in recent years more literature has become available and there is greater interest in muscular strength as another powerful marker of health, more information is needed to corroborate or refute our findings.

Flexibility is the least studied fitness component in connection with academic achievement. Our study showed that flexibility was positively associated with academic achievement. In this line, another study found that flexible children performed significantly better in mathematics and science [45]. Likewise, Bilgin et al. [42] found a positive relationship between flexibility and academic achievement in boys. Conversely, another study found no significant association with academic achievement [46]. The limited number of published studies together with the differences in the tests applied might explain such heterogeneous results.

Our results suggest that physical fitness and especially cardiorespiratory fitness play a potential role in intelligence and academic achievement. Therefore, physical activity interventions that promote improvements in fitness components could be very successful in enhancing academic achievement in the school setting [16]. Students should be given opportunities to engage in a wide variety of sports and physical activities, consistent with the idea that cardiorespiratory fitness can be developed and improved through different curricular content to attain greater student motivation and inclusion and high quality physical education [47]. Creating opportunities and environments to promote physical activity in students with low academic achievement could be a recommended strategy for schools to reduce school dropout, which constitutes a personal and social problem in Spain and is a European Union 2020 strategy [48]. Therefore, although there is no consensus of the physical activity characteristics for improving cognition or academic achievement, the global physical activity guidelines recommends moderate or vigorous PA for a minimum of 60 min daily that can be divided into 2 or more sessions, mostly aerobic and interspersing vigorous activities for muscle and bone strengthening 3 times a week [49]. However, further studies or meta-analyses should shed light on the potential physical activity prescription for improving brain health in adolescents [50].

Our findings support the argument for not reducing physical activity time and increasing high intensity activity time during or after school for its positive effects on brain health in students [51,52,53].

The main limitation of this study is its cross-sectional design, which cannot establish causality. Not including potential confounders such as body mass index or maturational status should also be noted as a limitation. It is important to highlight that the sample size decreased in general intelligence analyses, and therefore, we might be underpowered to detect associations between some of the fitness components and general intelligence. However, given the small size of the association between muscular and flexibility fitness with general intelligence, this finding will likely be non-significant even with a larger sample size. The main strength of this study is that, to the best of our knowledge, it is the first to examine the relationship between different physical fitness components and general intelligence using several reliable and valid tests. A further strength is the use of standardized and valid fitness tests.

## 5. Conclusions

Overall, our study showed a positive association between cardiovascular fitness and general intelligence. Further, cardiorespiratory fitness, muscular strength and flexibility were associated with academic achievement in adolescents. Our findings support the indication for increasing the time adolescents spend engaging in physical activities to improve components of fitness as a tool with the potential to positively impact intelligence and academic achievement. Further studies are needed to corroborate our findings.

## Figures and Tables

**Figure 1 ijerph-17-04362-f001:**
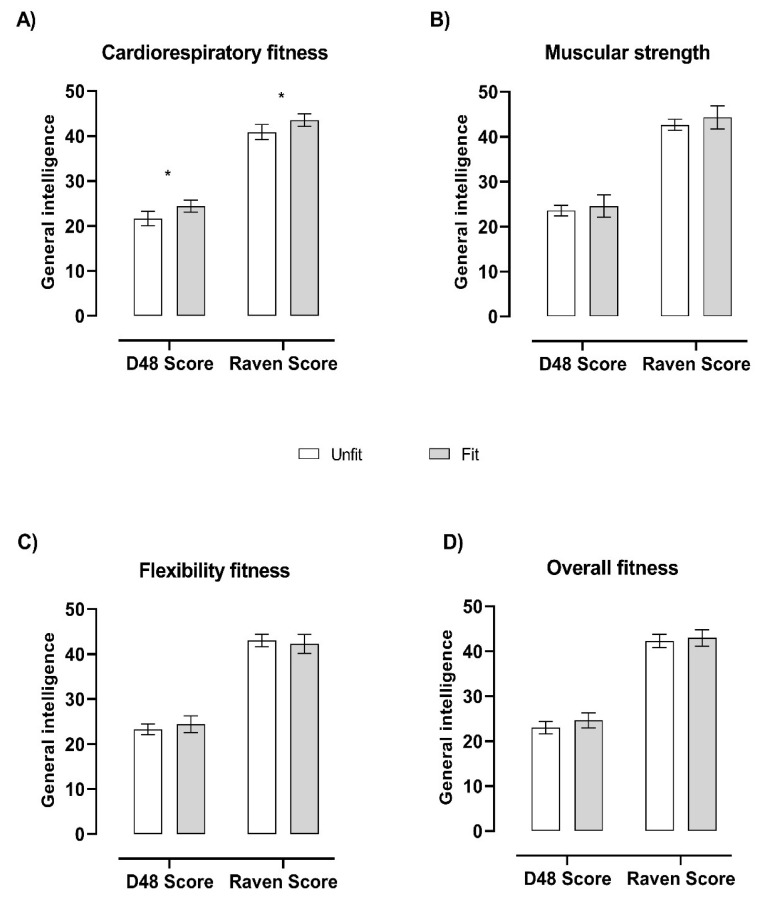
Physical fitness groups (i.e., unfit and fit) and general intelligence. The unfit and fit groups were classified according to the z-score for each fitness component (Figure 1A for cardiorespiratory fitness, Figure 1B for muscular strength, Figure 1C for flexibility fitness, and Figure 1D for overall composite z-score fitness). Sample size varies for the fitness components examined: cardiorespiratory fitness (unfit, *n* = 61; fit, *n* = 81), muscular strength (unfit, *n* = 111; fit, *n* = 25), flexibility (unfit, *n* = 97; fit, *n* = 43), and overall fitness (unfit, *n* = 79; fit, *n* = 51). Analyses of covariance (ANCOVA) adjusted for sex and age were performed to test differences between the unfit and fit groups. * *p* value < 0.01.

**Figure 2 ijerph-17-04362-f002:**
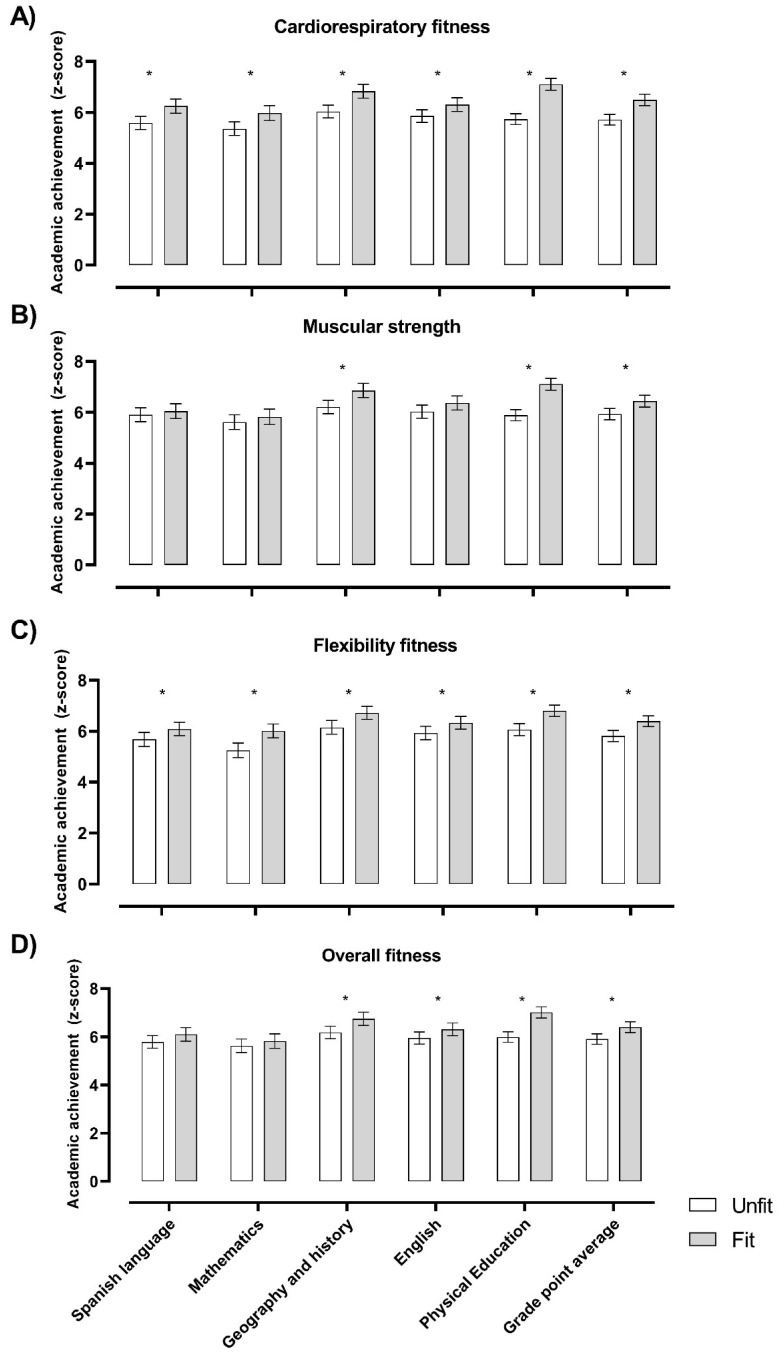
Physical fitness groups (i.e., unfit and fit) and level of academic achievement. The unfit and fit groups were classified according to the z-score for each fitness component (Figure 2A for cardiorespiratory fitness, Figure 2B for muscular strength, Figure 2C for flexibility fitness, and Figure 2D for overall composite z-score fitness). Sample size varies for the fitness components examined: cardiorespiratory fitness (unfit, *n* = 203; fit, *n* = 172), muscular strength (unfit, *n* = 199; fit, *n* = 175), flexibility (unfit, *n*= 180; fit, *n*= 200), and overall fitness (unfit, *n* = 191; fit, *n* = 170). Analyses of covariance (ANCOVA) adjusted for sex and age were performed to test differences between the unfit and fit groups. * *p* value < 0.05.

**Table 1 ijerph-17-04362-t001:** Descriptive characteristics of the study sample.

Characteristics	Total		Boys		Girls		*p* *
	*N*	Mean ± SD	*N*	Mean ± SD	*N*	Mean ± SD	
Age (years)	403	13.7 ± 1.2	216	13.7 ± 1.1	187	13.7 ± 1.2	0.822
Physical fitness:							
Cardiorespiratory fitness (VO_2_max **, mL/kg/min)	389	42.2 ± 6.3	208	44.6 ± 6.2	181	39.4 ± 5.3	<0.001
Muscular strength (cm)	396	162.3 ± 34.7	212	176.8 ± 35.0	184	145.6 ± 25.6	<0.001
Flexibility (cm)	403	2.7 ± 8.1	216	−0.01 ± 7.6	187	5.8 ± 7.7	<0.001
Overall fitness score (z-score)	369	0.0 ± 1.6	198	0.0 ± 1.7	171	0.0 ± 1.5	0.812
General intelligence:							
D48 (standard score)	161	23.2 ± 6.4	84	22.6 ± 5.8	77	23.8 ± 6.9	0.321
Raven (direct score)	160	42.5 ± 6.9	83	41.6 ± 7.1	77	43.5 ± 6.5	0.036
Academic Achievement (school grades, 0–10 points):							
Spanish Language	403	5.8 ± 1.9	216	5.5 ± 1.8	187	6.2 ± 1.9	0.001
Mathematics	403	5.6 ± 2.0	216	5.5 ± 2.0	187	5.7 ± 2.0	0.218
Geography and History	403	6.4 ± 1.8	216	6.2 ± 1.8	187	6.6 ± 1.9	0.052
English	403	6.1 ± 1.8	216	5.8 ± 1.8	187	6.3 ± 1.7	0.023
Physical Education	403	6.4 ± 1.6	216	6.3 ± 1.6	187	6.4 ± 1.6	0.425
Grade point average	403	6.0 ± 1.6	216	5.8 ± 1.6	187	6.2 ± 1.6	0.012

SD = Standard deviation. ** VO_2_max was calculated following the Léger et al. (1988) equation [22]. * *p* value refers to T-test sex-differences between boys and girls.

**Table 2 ijerph-17-04362-t002:** Linear regression analyses between physical fitness components and general intelligence and academic achievement.

Intelligence and Academic Achievement	Cardiorespiratory Fitness (VO_2_max Estimated *)				Muscular Strength (Standing Long Jump, cm)				Flexibility (Sit and Reach, cm)				Overall Fitness (z-Score)			
	β	B	95% CI	*p*	β	B	95% CI	*p*	β	B	95% CI	*p*	β	B	95% CI	*p*
General intelligence:																
D48 (standard score)	0.184	0.199	0.035–0.332	0.016	0.100	0.442	−0.270–1.155	0.222	0.031	0.026	−0.150–0.213	0.734	0.122	0.032	−0.014–0.079	0.168
Raven (direct score)	0.183	0.165	0.022–0.307	0.024	−0.110	−0.121	−0.283–0.040	0.139	0.116	0.476	−0.170–1.123	0.148	0.094	0.024	−0.020–0.067	0.282
Academic achievement:																
Spanish Language	0.124	0.416	0.121–0.712	0.006	0.049	0.902	−0.557–2.362	0.225	0.095	0.412	0.023–0.802	0.038	0.151	0.135	0.043–0.227	0.004
Mathematics	0.143	0.345	0.065–0.626	0.016	0.050	0.876	−0.513–2.264	0.216	0.159	0.652	0.289–1.015	<0.001	0.146	0.123	0.037–0.210	0.005
Geography and History	0.165	0.567	0.269–0.864	<0.001	0.172	3.273	1.815–4.731	<0.001	0.142	0.637	0.243–1.031	0.002	0.258	0.235	0.145–0.326	<0.001
English	0.095	0.333	0.026–0.641	0.034	0.089	1.731	0.212–3.249	0.026	0.115	0.531	0.124–0.938	0.011	0.158	0.148	0.053–0.244	0.002
Physical Education	0.320	1.225	0.91–1.536	<0.001	0.318	6.977	5.417–8.536	<0.001	0.229	1.184	0.745–1.624	<0.001	0.461	0.485	0.390–0.580	<0.001
Grade point average	0.189	0.773	0.420–1.127	<0.001	0.155	3.546	1.782–5.311	<0.001	0.175	0.943	0.471–1.415	<0.001	0.271	0.297	0.188–0.405	<0.001
Grade point average (without PE)	0.142	0.551	0.212–0.890	0.002	0.103	2.212	0.532–3.893	0.010	0.149	0.752	0.305–1.199	0.001	0.211	0.205	0.107–0.315	<0.001

**β** = Standardized beta coefficients. B = Unstandardized beta coefficients. CI = Confidence Interval. * VO_2_max was calculated following the Léger et al. (1988) equation [22]. Data were adjusted for sex and age.

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
