# Peer review of "Association of Physical Fitness with Intelligence and Academic Achievement in Adolescents"

_ijerph, 2020, doi:10.3390/ijerph17124362_

Round 1

Reviewer 1 Report

This cross-sectional study examined the relationship between physical fitness, cognition, and academic achievement in adolescents from the south of Spain. It showed a positive association between different components of physical fitness and general intelligence and academic achievement in adolescents.

Major point: 

Data collection took place during the first term of the 2015/2016 school year ??? what about up-to-date data research!!! 5 years old

Minor points:

Title: Please improve it. I suggest “Association of Physical fitness with

intelligence and academic achievement in adolescents”

Abstract: Please include the date/duration.

Please update references as some of them are old.

Author Response

Dear Editor,

Enclosed you will find a revision of our manuscript, “Association of physical fitness with intelligence and academic achievement in adolescents”. We would like to sincerely thank the Reviewers for their thoughtful and constructive comments. We have considered all of the suggestions and have incorporated them into the revised manuscript and we believe our manuscript is stronger as a result of these modifications. Changes to the original manuscript are highlighted in yellow and an itemized point-by-point response to the reviewers’ comments is presented below. 

COMMENTS FROM THE REVIEWER 1

Comments

Major point:

Data collection took place during the first term of the 2015/2016 school year ??? what about up-to-date data research!!! 5 years old

Minor points:

Title: Please improve it. I suggest “Association of physical fitness with intelligence and academic achievement in adolescents”

Abstract: Please include the date/duration.

Please update references as some of them are old.

Answer

Thank you for your comments.

Major point:

Thank you for your question. Although it may seem like five years, the total data collection was actually completed by the end of that school year, that is, September 2016. The inclusion of all the information in a matrix, the final cleaning of data and the first analyses took a time that placed us almost in 2018 when this article really started to be written and finally to leave it at the level that this journal deserves made it to the end of 2019 when the article was almost finished. Therefore, although a priori seems a long time from the beginning to the end of the process, as the reviewer and Editor know, all the process of a project behind needs even more time than the data collection itself. Furthermore, we strongly believe that the manuscript present novel and important results that will call the readers’ attention.

Minor points:

Thank you for the suggestions. We have incorporated all the changes suggested in title and abstract, as well as included updated references (see below the new papers included).

Page 1, abstract: We recruited 403 adolescents (53.6% boys) with a mean age of 13.7 ±1.2 years from a secondary school in Spain with a medium socioeconomic status, during 2015/2016 school year

References:

Page 17: Álvarez-Bueno, C.; Hillman, C. H.; Cavero-Redondo, I.; Sánchez-López, M.; Pozuelo-Carrascosa, D. P.; Martínez-Vizcaíno, V. Aerobic Fitness and Academic Achievement: A Systematic Review and Meta-Analysis. J. Sports Sci., 2020. https://doi.org/10.1080/02640414.2020.1720496.

Page 18: Department of Health & Human Services. Physical Activity Guidelines Advisory Committee. Phys. Act. Guidel. Advis. Comm. Sci. Rep., 2018, 779. https://doi.org/10.1111/j.1753-4887.2008.00136.x.

Reviewer 2 Report

REVIEW

Manuscript number ijerph-814049

Title: "Physical fitness is positively associated with intelligence and academic achievement in adolescents"

GENERAL COMMENTS:

The manuscript addresses important issue of physical fitness and its importance for intelligence and academic achievement. The results of the study are important as they might have an impact on decisions regarding the amount of hours of PE classes in schools. The impact of cardiorespiratory fitness on intelligence and academic achievements point out the need to offer children as many possibilities as possible in school for physical activity.

I have some additional comments and suggestions:

Title

I would suggest changing the title because the authors in the results stated that ‘muscular strength, flexibility and overall fitness were not associated with intelligence’. The results do not confirm what is stated in the title.

Abstract

The conclusion in the abstract does not match the results. It is the same case as with the title. I would suggest to change the conclusion to be more precise.

Materials and Methods

I would also suggest to add the description of the procedure like what was the order of the tests, were they done during one day? It would allow other researchers to replicate the study.

As for measurements it is not clear why authors decided to choose those 3 tests from EUROFIT. It should be explained in the methods section. Maybe balance or tapping which are also part of EUROFIT would be more suitable to compare them with intelligence or academic achievements.

Line 127

The long jump test is to jump as far as possible not as high. Please correct the description.

Results

Table 1

The characteristic of the participants is not sufficient as there is no information about body height and weight. This has to be included in the table.

Based on the information in the table only 160 participants completed the intelligence examination. What was the reason for reducing the amount of the participants?

There is no information in the paper how many participants were in the fit/unfit group based on each component.

The text describing figure 2 is repetitive of what can be seen on the figure. I would suggest to point out only the most important findings.

Discussion

I would suggest to be more precise in the first sentence of the discussion as based on the results of the study there only association between cardiorespiratory fitness and intelligence.

Conclusions

‘our study showed a positive association between different components of physical fitness and general intelligence’

The conclusions are being overinterpreted and are not supported by the results. I would suggest to be more precise as only cardiorespiratory fitness showed this association.

Minor comments

  • Figure 1 section D has to corrected in terms of colors
  • There are some spelling mistakes in the text
  • The line numbering end on the description of table 2

Author Response

Dear Editor,

Enclosed you will find a revision of our manuscript, “Association of physical fitness with intelligence and academic achievement in adolescents”. We would like to sincerely thank the Reviewers for their thoughtful and constructive comments. We have considered all of the suggestions and have incorporated them into the revised manuscript and we believe our manuscript is stronger as a result of these modifications. Changes to the original manuscript are highlighted in yellow and an itemized point-by-point response to the reviewers’ comments is presented below. 

COMMENTS FROM THE REVIEWER 2

Comment

Title: "Physical fitness is positively associated with intelligence and academic achievement in adolescents"

 GENERAL COMMENTS:

The manuscript addresses important issue of physical fitness and its importance for intelligence and academic achievement. The results of the study are important as they might have an impact on decisions regarding the amount of hours of PE classes in schools. The impact of cardiorespiratory fitness on intelligence and academic achievements point out the need to offer children as many possibilities as possible in school for physical activity.

 I have some additional comments and suggestions:

 Title

I would suggest changing the title because the authors in the results stated that ‘muscular strength, flexibility and overall fitness were not associated with intelligence’. The results do not confirm what is stated in the title.

Answer

Thank you for your comments and suggestion. We agree with the reviewer, and the title has now changed according to Reviewer 1 suggestion into “Association of physical fitness with intelligence and academic achievement in adolescents”.

Comment

Abstract

The conclusion in the abstract does not match the results. It is the same case as with the title. I would suggest to change the conclusion to be more precise.

Answer

Thank you for your suggestion. We have changed the conclusion, stating a more precise conclusion regarding our findings.

Page 1 and 13: Overall, cardiorespiratory fitness was positively associated with intelligence and academic achievement.

Comment

Materials and Methods

I would also suggest to add the description of the procedure like what was the order of the tests, were they done during one day? It would allow other researchers to replicate the study.

As for measurements it is not clear why authors decided to choose those 3 tests from EUROFIT. It should be explained in the methods section. Maybe balance or tapping which are also part of EUROFIT would be more suitable to compare them with intelligence or academic achievements.

Answer

Thank you for your suggestion. We have included more information regarding the procedure followed during the assessment as required.

Page 3. Assessments were done during physical education classes in two different days: first day, the 20-m shuttle run test was performed, and the second day, we performed the standing long jump and sit-and-reach tests.

Moreover, as the assessment were done in a school context, and based on the previous evidence and most fitness component assessed, for feasibility reasons we decided for cardiorespiratory fitness test such as the most well-known and used test in children (i.e., 20m shuttle run test); for muscular strength, we measured standing long jump, and finally for flexibility component we performed the sit and reach test. These tests are also known for being feasible, reliable and valid in this population (1-4). We have included a sentence regarding the test selection.

Page 3. For feasibility reasons, we did not apply the full EUROFIT fitness test battery.

Comment

Line 127

The long jump test is to jump as far as possible not as high. Please correct the description.

Answer

Thank you for catching this error. We have corrected accordingly.

Page 4: Briefly, the participant had to jump as far as possible, remaining upright and using both feet.

Comment

Results

Table 1

The characteristic of the participants is not sufficient as there is no information about body height and weight. This has to be included in the table.

Answer

Thank you for your suggestion. As we have acknowledged in the limitation section, we did not measure height and weight in this study. However, based on the studies published in the last years on the topic, the body composition parameters are only used as descriptive indicator (5-9). Nevertheless, the physical fitness tests used are weight-bearing (weight-dependent tests), and therefore, the weight is controlling the results obtained.

Page 13: Not including potential confounders such as body mass index or maturational status should also be noted as a limitation.

Comment

Based on the information in the table only 160 participants completed the intelligence examination. What was the reason for reducing the amount of the participants?

Answer

Thank you for this suggestion. We have included in the results section a brief paragraph explaining the sample size reduction.

Page 6. To note that sample size decreases in some variables due to the missing data (i.e., did not attend to school the day of the evaluation). Specially, the general intelligence evaluation was decreased up to 161 participants, only including those participants that were engaged at the first level of the Spanish educational system.

Comment

There is no information in the paper how many participants were in the fit/unfit group based on each component.

Answer

Thank you for catching this. We have incorporated in the footnotes of the figures the sample size of fit/unfit group.

Page 9. Figure 1, footnote. Sample size varies for the fitness components examined: cardiorespiratory fitness (unfit, n= 61; fit, n= 81), muscular strength (unfit, n= 111; fit, n= 25), flexibility (unfit, n= 97; fit, n= 43), and overall fitness (unfit, n= 79; fit, n= 51).

Page 10. Figure 2, footnote. Sample size varies for the fitness components examined: cardiorespiratory fitness (unfit, n= 203; fit, n= 172), muscular strength (unfit, n= 199; fit, n= 175), flexibility (unfit, n= 180; fit, n= 200), and overall fitness (unfit, n= 191; fit, n= 170).

Comment

The text describing figure 2 is repetitive of what can be seen on the figure. I would suggest to point out only the most important findings.

Answer

Thank you for your suggestion. We have reduced the information provided regarding the Figure 2, pointing out an overall important finding.

Page 10. Figure 2 shows the relationship between the unfit and fit groups and academic achievement outcomes. Overall, students classified as fit presented better academic achievement than those in the unfit group.

Comment

Discussion

I would suggest to be more precise in the first sentence of the discussion as based on the results of the study there only association between cardiorespiratory fitness and intelligence.

Answer

Thank you for your suggestion. As we have done previously, we have been consistently in the idea of the association of cardiorespiratory fitness with intelligence.

Page 11. The main findings of our study suggest that cardiorespiratory fitness is associated with general intelligence. Further, cardiorespiratory fitness, muscular strength and flexibility are associated with academic achievement in adolescents.

Comment

Conclusions

‘our study showed a positive association between different components of physical fitness and general intelligence’

The conclusions are being overinterpreted and are not supported by the results. I would suggest to be more precise as only cardiorespiratory fitness showed this association.

Answer

Thank you for your comment. We have rephrased the conclusion section accordingly.

Page 13. Overall, our study showed a positive association between cardiovascular fitness and general intelligence. Further, cardiorespiratory fitness, muscular strength and flexibility were associated with academic achievement in adolescents.

Comment

Minor comments

  • Figure 1 section D has to corrected in terms of colors
  • There are some spelling mistakes in the text
  • The line numbering end on the description of table 2

Answer

Thank you for catching these typo errors. We have corrected them and reviewed all the document accordingly. In regards to the line numbering, in our submission document we do not include line numbering according to the Journal rules. Then, the submission system includes the line numbering automatically.

REFERENCES

  1. Léger LA, Mercier D, Gadoury C, Lambert J. The multistage 20 metre shuttle run test for aerobic fitness. J Sports Sci. 1988;6(2):93‐101. doi:10.1080/02640418808729800
  2. Fernandez-Santos JR, Ruiz JR, Cohen DD, Gonzalez-Montesinos JL, Castro-Piñero J. Reliability and Validity of Tests to Assess Lower-Body Muscular Power in Children. J Strength Cond Res. 2015;29(8):2277‐2285. doi:10.1519/JSC.0000000000000864
  3. Patterson P, Wiksten DL, Ray L, Flanders C, Sanphy D. The validity and reliability of the back saver sit-and-reach test in middle school girls and boys. Res Q Exerc Sport. 1996;67(4):448‐451. doi:10.1080/02701367.1996.10607976
  4. Ruiz JR, Castro-Pinero J, Espana-Romero V et al. Field-based fitness assessmentin young people: the ALPHA health-related fitness test battery for children andadolescents. Br J Sports Med 2010; 45(6):518–524.17
  5. Borkertienė, V.; Stasiulis, A.; Zacharienė, B.; Kyguolienė, L.; Bacevičienė, R. Association among Executive Function, Physical Activity, and Weight Status in Youth. Medicina 2019, 55, 677.
  6. Jirout, J.; LoCasale-Crouch, J.; Turnbull, K.; Gu, Y.; Cubides, M.; Garzione, S.; Evans, T.M.; Weltman, A.L.; Kranz, S. How Lifestyle Factors Affect Cognitive and Executive Function and the Ability to Learn in Children. Nutrients 2019, 11, 1953.
  7. Reigal, R.E.; Moral-Campillo, L.; Morillo-Baro, J.P.; Juárez-Ruiz de Mier, R.; Hernández-Mendo, A.; Morales-Sánchez, V. Physical Exercise, Fitness, Cognitive Functioning, and Psychosocial Variables in an Adolescent Sample. Int. J. Environ. Res. Public Health2020, 17, 1100.
  8. Moradi, A.; Sadri Damirchi, E.; Narimani, M.; Esmaeilzadeh, S.; Dziembowska, I.; Azevedo, L.B.; Luiz do Prado, W. Association between Physical and Motor Fitness with Cognition in Children. Medicina 2019, 55, 7.
  9. Bleiweiss-Sande, R.; Chui, K.; Wright, C.; Amin, S.; Anzman-Frasca, S.; Sacheck, J.M. Associations between Food Group Intake, Cognition, and Academic Achievement in Elementary Schoolchildren. Nutrients 2019, 11, 2722.

Reviewer 3 Report

This is a cross-sectional study examining the correlation between physical fitness and intelligence and academic achievement among adolescents. The study included 403 adolescents from a secondary school. Physical fitness, including cardiorespiratory fitness, muscular strength, and flexibility were measured. General intelligence was measured using D48 and Raven’s Progressive Matrices. Academic achievement was assessed using the grades of each subject in the first term of the assessment year. using liner regression models, the study found that physical fitness, especially cardiorespiratory fitness, was positively associated with intelligence and academic achievement after adjusting for age and sex. The manuscript is well written overall.

I only have one question: you mentioned that socioeconomic status (e.g., family income of each participant) was collected in the study; is it a potential confounder of the associations between physical fitness and intelligence/academic achievement?

Author Response

Dear Editor,

Enclosed you will find a revision of our manuscript, “Association of physical fitness with intelligence and academic achievement in adolescents”. We would like to sincerely thank the Reviewers for their thoughtful and constructive comments. We have considered all of the suggestions and have incorporated them into the revised manuscript and we believe our manuscript is stronger as a result of these modifications. Changes to the original manuscript are highlighted in yellow and an itemized point-by-point response to the reviewers’ comments is presented below. 

COMMENTS FROM THE REVIEWER 3

Comment

This is a cross-sectional study examining the correlation between physical fitness and intelligence and academic achievement among adolescents. The study included 403 adolescents from a secondary school. Physical fitness, including cardiorespiratory fitness, muscular strength, and flexibility were measured. General intelligence was measured using D48 and Raven’s Progressive Matrices. Academic achievement was assessed using the grades of each subject in the first term of the assessment year. using liner regression models, the study found that physical fitness, especially cardiorespiratory fitness, was positively associated with intelligence and academic achievement after adjusting for age and sex. The manuscript is well written overall.

I only have one question: you mentioned that socioeconomic status (e.g., family income of each participant) was collected in the study; is it a potential confounder of the associations between physical fitness and intelligence/academic achievement?

Answer

Thank you very much for your comment. The medium socioeconomic status was measure of the school based on the Andalusian Educational Evaluation Agency report (level five out of ten), according to items including family income, parents’ profession, parents’ educational level and facilities for studying at home. Therefore, all the participants selected were grouped in the same socioeconomic status and, therefore, there is no need to be included as a confounder.

Reviewer 4 Report

Dear authors please find attached some minor revisions and a few major issues. 

TITLE/ABSTRACT/REFERENCES

Title and abstract give an overall view of the paper features, well-identifying the objectives of the present work. The abstract correctly describes the population, the methods used to characterise the physical fitness components (namely cardiorespiratory fitness, muscular strength and flexibility) as well as the intelligence and the academic achievements.

However, none statistical methods are mentioned. Please provide the methods and also the interval confidences boundaries. Could be appropriate rounding the reported age digits (13.73) to three or even fewer digits (i.e. 13.7), which befits better to the sense of the measure itself (not only in the abstract).

The references are appropriate.

METHODS

  1. Please provide more details on the subjects’ selection, it is not clear how the authors reached a sample size of 403 subjects from the original project. How many subjects were included in the original project? Why and how were 403 subjects selected? (randomly? Complete-case analysis? Etc.)
  2. Was this sample size appropriate? (Please state if any sample size calculation has been made)
  3. Please provide a general statement on the selected type I error and the ICs width.
  4. (line 130) Please define how the flexibility measurement was expressed (centimetres?)
  5. (line 151) what measure did the authors take for the analysis concerning Raven’s test results?
  6. (line 163) The authors stated that they used t-test to determine whether there were differences in the mean between boys and girls. Was this test appropriate? Were the analysed variables normally (or approximately) distributed? Did the authors check for normality?
  7. (line 169) The authors stated that they tested the association between the intelligence and the physical fitness components by a linear regression model. Did they previously identify a linear trend? If yes, how was it cheeked? Moreover, were the linear regression assumptions verified?

RESULTS

  1. Table 1: please state that some variables contain missing data, it has been specified nowhere (e.g. muscular strength n= 396 out of 403, overall fitness score n = 369 out of 403, D48 standard score n = 161 out of 403 and Raven 160 out of 403). In addition, the sum of boys and girls does not correspond to the total (cardiorespiratory fitness total = 389, boys + girls = 228+195 = 423!).
  2. Please provide the frequencies too. Boys ‘flexibility means is reported as -0.01 cm, is that possible? It could be worthy of reporting flexibility as median ± IQR due to its large variability. Please add details on the statistical test used (t-test?) in the table caption/footnote.
  3. Table 2: it is not clear if the authors performed four separated LRMs (one each DV) testing for all IVs at the same time. Please provide additional details in the statistical method paragraph and better state if the analyses were performed separately. Some IVs may be correlated, did the author checked for collinearity?
  4. Figure 1 reports additional results about fit/unfit group comparisons. Please add details on the group’s numerousness and on the used statistical test, since each table/figure should be considered as stand-alone. Consider adding a short paragraph/statement about these analyses.

The main concern is about the missing data that affected the LRM(s), which, I guess, had only 159 degrees of freedom and not 402. How did the authors deal with the missing data to solve this problem?

Reviewer 5 Report

  1. Please provide appropriate references for the 20m shuttle run test.
  2. Provide appropriate references to the sit-and -reach test.
  3. As part of the introduction / literature review and discussion can the authors indicate what is the recommended frequency, duration for adolescent to engage in physical activities to potentially achieve a significant improvement in academic and cognitive scores. 

Author Response

Dear Editor,

Enclosed you will find a revision of our manuscript, “Association of physical fitness with intelligence and academic achievement in adolescents”. We would like to sincerely thank the Reviewers for their thoughtful and constructive comments. We have considered all of the suggestions and have incorporated them into the revised manuscript and we believe our manuscript is stronger as a result of these modifications. Changes to the original manuscript are highlighted in yellow and an itemized point-by-point response to the reviewers’ comments is presented below. 

COMMENTS FROM THE REVIEWER 5

Comment

  1. Please provide appropriate references for the 20m shuttle run test.

Answer

Thank you for your comment. We already have incorporated the appropriate reference for the 20m shuttle run test, and the equation provided for the estimation of the VO2max.

Reference: Léger, L. A.; Mercier, D.; Gadoury, C.; Lambert, J. The Multistage 20 Metre Shuttle Run Test for Aerobic Fitness. J. Sports Sci., 1988, 6 (2), 93–101. https://doi.org/10.1080/02640418808729800.

Comment

  1. Provide appropriate references to the sit-and -reach test.

Answer

Thank you. We have included the following reference in regard to sit-and reach test:

Reference: Adam C, Klissouras V, Ravazzolo M, Renson R, Tuxworth W, Kemper HCG et al. EUROFIT - European test of physical fitness (2nd edition). Council of Europe. Committee for the development of sport. 2 ed. Strasbourgh: Council of Europe, 1993.

Comment

  1. As part of the introduction / literature review and discussion can the authors indicate what is the recommended frequency, duration for adolescent to engage in physical activities to potentially achieve a significant improvement in academic and cognitive scores.

Answer

Thank you for your suggestion. We agree with the reviewer and we have included the following information in the discussion.

Page 13. Therefore, although there is no consensus of the physical activity characteristics for improving cognition or academic achievement, the global physical activity guidelines recommends moderate or vigorous PA for a minimum of 60 min daily, that can be divided into 2 or more sessions, mostly aerobic and interspersing vigorous activities for muscle and bone strengthening 3 times a week [50]. However, furthers studies or meta-analysis should shed the light on the potential physical activity prescription for improving brain health in adolescents [51].

Round 2

Reviewer 1 Report

Dear Authors,

Thank you for considering the suggestions.

I believe the manuscript is stronger now.

Many thanks

Reviewer 4 Report

Dear authors,

thank you very much for your great job. I think the manuscript has improved a lot.

Best regards.